# Temporal Muscle Thickness Compared to Functional Scales as a Prognostic Parameter in Patients with Brain Metastases

**DOI:** 10.3390/cancers16091660

**Published:** 2024-04-25

**Authors:** Julia Klingenschmid, Aleksandrs Krigers, Daniel Pinggera, Johannes Kerschbaumer, Nadine Pichler, Victoria Schoen, Matthias Demetz, Astrid E. Grams, Claudius Thomé, Christian F. Freyschlag

**Affiliations:** 1Department of Neurosurgery, Medical University of Innsbruck, Anichstrasse 35, 6020 Innsbruck, Austria; ju.klingenschmid@tirol-kliniken.at (J.K.);; 2Department of Radiology, Medical University of Innsbruck, Anichstrasse 35, 6020 Innsbruck, Austria

**Keywords:** frailty, temporal muscle thickness, brain metastases, neurosurgery, prognosis

## Abstract

**Simple Summary:**

Temporal muscle thickness has become an interesting parameter in recent neuro-oncological research due to promising results regarding the estimation of patients’ prognosis. The aim of our study was to assess its usefulness in terms of estimating prognosis after the surgical resection of brain metastases compared to functional scales. This research did not confirm previously published data postulating a significant influence of thicker temporal muscle on patient survival, but proved no association with better outcomes after neurosurgical therapy.

**Abstract:**

Metastases are the most frequent intracranial malignant tumors in adults. While Karnofsky Performance Status (KPS) and Clinical Frailty Scale (CFS) are known to have significant impact on overall survival (OS), temporal muscle thickness (TMT) has been postulated to be a promising new parameter to estimate prognosis. Patients who received a resection of one to three brain metastases in our institution were included. Temporal muscle thickness was measured in preoperative MRI scans according to a standardized protocol. In 199 patients, the mean TMT was 7.5 mm (95CI 7.3–7.7) and the mean OS during follow-up was 31.3 months (95CI 24.2–38.3). There was no significant correlation of TMT and preoperative or follow-up CFS and KPS. While CFS and KPS did significantly correlate with OS (*p* < 0.001 for each), no correlation was demonstrated for TMT. CFS showed a superior prognostic value compared to KPS. TMT failed to show a significant impact on OS or patient performance, whereas the clinical scales (KPS and CFS) demonstrate a good correlation with OS. Due to its superiority over KPS, we strongly recommend the use of CFS to estimate OS in patients with brain metastases.

## 1. Introduction

Neurosurgical therapy for patients suffering from brain tumors has not been restricted to the young and healthy population but has continuously been adapted and applied to patients with all kinds of conditions and comorbidities. Not only cancer but also various other medical conditions result in increased patient frailty–characterized by reduced physiologic function and diminished strength and endurance. Frail patients develop a higher vulnerability to being dependent and, further, an increased mortality [1].

Previous research demonstrated that frailty has a significant impact on overall survival (OS) in patients with brain metastases, resulting in poorer functionality and shorter OS [2]. To assess functionality or frailty, there are several scales and methods which characterize the performance levels of our patients. In metastasized malignant disease, sarcopenia could be an objective diagnosis to estimate the systemic tumor burden. In recent neuro-oncological research, the measurement of temporal muscle thickness (TMT) has been suggested as a useful prognostic marker for OS in patients with brain metastases of breast cancer, melanoma, and non-small-cell lung cancer (NSCLC) [3,4,5,6]. In neuro-oncological patients, for example, patients with brain metastases, clinical visits are usually accompanied by cranial magnetic resonance imaging (MRI), offering the opportunity to quantitatively assess sarcopenia using TMT as a direct surrogate to assess skeletal muscle mass (SMM) [7,8,9,10]. A scoring system to quantify frailty is the Clinical Frailty Scale (CFS) by Rockwood et al., with its updated version (CFS 2.0) published in 2020 (Figure 1) [11]. It includes a variety of physical and mental health factors and allows doctors to assess patients and easily categorize them into one of nine groups, ranging from “very fit” to “terminally ill”. Initially, CFS was implemented in geriatric medicine where the main contributing factors, in addition to sarcopenia, are abnormal endocrine and inflammatory function as well as poor energy regulation [12,13]. Furthermore, the scoring system has been introduced to other medical specialties, such as intensive care medicine, where it predicts long-term mortality [14]. At the same time, CFS started to play a role in operative disciplines, forecasting revision surgeries and associated morbidity in orthopedic surgery [15]. CFS recently proved to be a reliable tool to predict OS in neuro-oncological patients and has shown superiority to the Karnofsky Performance Status Scale (KPS) in patients with high-grade glioma and brain metastases [2,16].

In glioblastoma patients, various studies suggested TMT to be a useful parameter for OS because of significantly better OS with a thicker TMT [17,18]. Recent investigations, however, contradicted these results by showing no influence of TMT on OS [19,20,21]. So far, data for TMT in patients with brain metastases are lacking, although TMT could be a promising independent marker for OS in these patients.

The objective of this study was to determine whether TMT as a quick and radiologically easily applicable parameter is of additional usefulness for the clinical assessment of patients’ frailty (using CFS) in predicting OS in patients with brain metastases before and after undergoing neurosurgical resection. Further, we aimed to identify realistic cut-off values based on measurements of patients harboring metastases in our neuro-oncological database.

**Figure 1 cancers-16-01660-f001:**
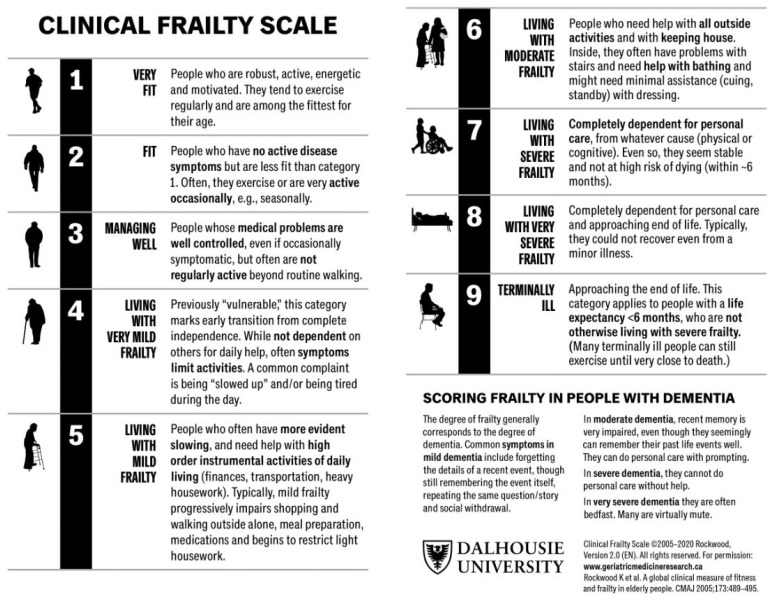
The updated Clinical Frailty Scale by Rockwood et al. divides patients into nine groups according to their grade of frailty, including factors such as physical activity, independence, and disease symptoms [22]. www.geriatricmedicineresearch.ca, accessed on 1 December 2023.

## 2. Materials and Methods

Patients undergoing a resection of 1–3 brain metastases in our institution were included. Surgical decisions were taken according to the interdisciplinary neuro-oncology tumor board. Only surgical patients were included. In a retrospective approach, TMT was measured using previously published protocols [4]: in the preoperative MRI scan, measurements were taken in axial images of contrast-enhanced T1-weighted scans. The orbital roof served as a landmark for cranio-caudal orientation, which was verified in the sagittal plane. For fronto-occipital orientation, the Sylvian fissure was used. There, we measured the temporal muscle from the inner to the outer margin, whilst its fascia was meticulously excluded from measurements. Bilateral measurements of TMT were performed and the mean was calculated. In postoperative measurements of patients with a history of surgical temporal muscle dissection, we performed exclusively unilateral measurements of the non-dissected side. An example of a TMT measurement is shown in Figure 2.

Each patient’s CFS and KPS were retrospectively analyzed by reviewing patients’ charts from the preoperative visit, postoperatively before patient discharge, and at a first follow-up visit three to six months after surgery. Our neuro-oncological database additionally provided epidemiological and neuropathological data as well as the number of metastases, the origin of primary tumor, and the extent of resection according to the surgeon’s estimation and to postoperative MRI scans.

This study was approved by the Ethics Committee of the Medical University of Innsbruck (1333/2021) and conducted according to the ethical standards of the 1975 Declaration of Helsinki, amended in 2013.

Statistical analysis: IBM SPSS Statistics (IBM SPSS Statistics for Mac OS, Version 27.0. Armonk, NY: IBM Corp.) was used for processing statistical analysis and graphs. Correlations were detected using Pearson analysis. A *T*-test supported monovariate analysis and the influence of multiple variables was assessed using linear regression. OS was assessed using Kaplan–Meier processing with log-rank test. Cox regression was used for calculating death hazard ratios for independent parameters. The TMT cut-off value was determined by means of receiver operating characteristic (ROC) analysis and area under the curve (AUC) with subsequent Youden Index assessment. *p* values < 0.05 were considered statistically significant.

## 3. Results

### 3.1. Cohort Description

This study included a total of 199 patients, consisting of 106 (53.3%) men and 93 (46.7%) women. Measurements in this study cohort showed a mean TMT of 7.8 mm (95CI 7.4–8.2) in male and 7.3 mm (95CI 6.9–7.7) in female patients, and the overall mean TMT was 7.5 mm (95CI 7.3–7.8). The mean preoperative CFS was 3 (95CI 3.1–3.4), while the mean preoperative KPS was 80 (95CI 73–86). In the follow-up visit, patients scored a mean CFS of 3 (95CI 2.6–3.2) and a mean KPS of 80 (95CI 81.8–87.4). The mean age in this cohort was 60 years (95CI 58.8–62.1). The most common primary tumor was NSCLC, followed by breast and colorectal cancer (Table 1).

In 86.9% (*n* = 173) of cases, a singular metastasis was resected, whereas in 12.6% (*n* = 25), a resection of two metastases was performed, and in one case, three metastases were removed. A total of 82.4% of all metastases were located supratentorially and 17.6% infratentorially; 25.1% were in eloquent brain areas and 74.9% in areas without eloquence.

During the follow-up period of this study, 19% of patients showed a local progression, 30.3% developed distant cerebral progression, and 40.5% showed extra-cerebral progression of the disease. A total of 10.2% showed no ongoing disease. Following resection, 46.2% received whole brain radiotherapy, 25.9% were radiated locally, and in 26.4%, no postoperative radiotherapy was performed. In 1.5%, there were no data about adjuvant radiotherapy.

### 3.2. TMT and Functional Scores

TMT measurements obtained in preoperative MRI scans demonstrated no significant correlation with KPS or CFS in preoperative, postoperative, or follow-up assessments. Detailed results are depicted in Table 2 and Table 3.

### 3.3. Interrelations of TMT

No significant correlation of TMT with the number of metastases, metastasis volume, resected volume, edema volume, or functional status assessed with KPS and CFS could be demonstrated (*p* = n.s.). Postoperative functional scores, however, demonstrated a correlation of a thinner TMT with poorer KPS and CFS (r = 0.188, *p* < 0.01 for KPS, r = −0.158, *p* < 0.05 for CFS). Further, there was no correlation of TMT with local, distant, or systemic tumor progression after surgery (*p* = n.s.). Only female gender and metastasis location in the left hemisphere showed a significant correlation with a thinner TMT (*p* < 0.05 each). 

### 3.4. Overall Survival (OS)

The mean OS in the follow-up period of this study was 31.3 months (95CI 24.2–38.3). In Cox regression, TMT showed no significant impact on OS (*p* = n.s.). Regarding functional scores at the time of TMT measurement, however, a reduction in OS by 30% (95CI 16.3–43.8, *p* < 0.001) per 10 units’ deficit in KPS and a reduction by 35% (95CI 1.19–1.52, *p* < 0.001) per increasing step in CFS score was demonstrated. Age appeared to have a significant impact on OS by raising the hazard of death within our follow-up by 2.4% (95CI 1.01–1.04, *p* < 0.01) per increased year of patients’ age.

### 3.5. ROC/AUC Data

No significant AUC values could be demonstrated for TMT measurements regarding 6-, 12-, and 24-month OS (*p* = n.s.). Preoperative functional scores showed significant results for 6-month (AUC = 0.617, *p* < 0.01) and 12-month (AUC = 0.640, *p* < 0.01) OS regarding KPS. Analogically, CFS showed significant values in 6- (AUC = 0.615, *p* < 0.01), 12- (AUC = 0.649, *p* < 0.001), and 24-month (AUC = 0.592, *p* < 0.05) OS. Additionally, patient age appeared to have significant influence on OS greater than 6 and 12 months (AUC = 0.592 and 0.569, *p* < 0.05 for each). The respective graphs are shown in Figure 3. An optimal TMT cut-off value to estimate its influence on OS, using this study’s ROC data, was set at 5.45 mm (*p* = n.s.) for OS > 6 months, 6 mm (*p* = n.s.) for OS > 12 months, and 5.83 mm (*p* = n.s.) for OS > 24 months.

## 4. Discussion

The predictive value of TMT on OS in cancer patients with brain metastases has been described very promisingly in the existing literature, which inspired this investigation of patients in our institutional database and the comparison of TMT with CFS, another, already established, prognostic marker of OS in these patients. Unexpectedly, the results contradict many previously published results.

This study could not prove any predictive value of TMT on OS in patients with brain metastases either in Cox regression or in ROC analysis. Additionally, no significant association of TMT with functional scores (CFS, KPS) assessed before surgery and at follow-up could be confirmed. At the same time, CFS and KPS proved to have a significant correlation with OS depicting the probability to die within our follow-up period raised by 30% per worsened step in preoperative KPS and by 35% per worsened step in preoperative CFS. In long-term follow-up, CFS showed a significant effect on OS longer than six, twelve, and twenty-four months, whereas KPS only showed significant AUC values in OS greater than six and twelve months, indicating the superior prognostic value of CFS. 

While preoperative and follow-up functional scores demonstrated no correlation with TMT, postoperative scores assessed before patient discharge did significantly correlate with TMT, suggesting a higher perioperative vulnerability of patients with sarcopenia. This complements the literature stating that patients with sarcopenia are more prone to postoperative complications and have a higher in-hospital mortality rate [23]. While TMT cut-off thresholds for longer OS were recommended at 7.0 mm for female and 7.7 mm for male patients in one study [3], the preoperative risk for sarcopenia and subsequently increased risk of death were defined at TMT ≤ 6.3 mm in male patients and ≤5.2 mm in female patients and TMT < 11 mm for both genders by other groups [6,24]. Recommended TMT cut-off values strongly vary in the literature. Our findings somewhat agreed with one of the studies [24]; however, no significant results for the cut-off values were achieved. We tried to find an impact of lower TMT on OS in different follow-up periods (>6, >12, and >24 months), but could not detect any significant effect. In the literature, cut-off values have been inconsistent and therefore cannot be uniformly applied. Thus, no meaningful recommendation for a reasonable TMT cut-off value can be made at this point of national and international research.

Measurements of TMT in preoperative MRI scans can be elicited quite quickly and effortlessly if the image viewing software used provides a respective measuring tool. However, there remains space for errors while doing so, depending on the quality and thickness of MRI slices and also the preciseness of the professional performing the measurement. Preoperative CFS, on the other hand, is a self-explaining scale supported by eye-catching pictograms, which make a fitting assignation to one of the nine possible groups and the subsequent quantification of patients’ frailty extraordinarily easy. The assessment can be performed anywhere and does not require any hardware or technical skills; therefore, it can be performed not only by clinical professionals but also by other healthcare workers, students, or even the patients themselves.

Female gender was significantly associated with a thinner TMT. This coincides with previously conducted studies on TMT in high-grade gliomas [21] and in healthy individuals, where men were proven to have physiologically thicker TMT than women [25,26]. However, this does not equally apply at every age: the most remarkable gender-related difference can be found early in life, where TMT growth in male individuals accelerates at the onset of puberty, reaching a plateau in their early twenties [27], potentially because of testosterone-associated effects on the growth of muscle mass [28]. A study on normal values of TMT in the Japanese population showed no significant differences of TMT in patients by the age of seventy and above [26].

There are several factors and circumstances that might have an effect on TMT apart from sarcopenia and chronic disease. While malnutrition and a loss of body weight could decrease TMT as they lead to a decrease of general muscle mass, excessive chewing and different food preferences have not yet been investigated regarding their impact on TMT. Increased masticatory and dietary demands during childhood result in altered horizontal and vertical muscle growth [29], which has an effect on muscle thickness due to different dietary habits. Some cases of stress hypertrophy affecting the temporal muscle have been described [30].

How a left-hemisphere tumor location is linked to a thinner TMT remains unclear. A more rapid decrease in general physical activity with commonly dominant hemisphere involvement, possibly due to aphasia and consecutive social withdrawal, can be postulated; further studies are needed to investigate this assumption.

Even though metastatic cancer disease obviously has an effect on patients’ performance, sarcopenic patients (given the threshold of CFS > 4 defining a patient as frail) are a minority in neurosurgery, where those patients who are usually admitted for surgical treatment represent the best subpopulation of cancer patients.

## 5. Conclusions

TMT did not prove any usefulness in the prognosis of OS. Measuring TMT in preoperative MRI scans is easily conductible in neuro-oncological patients, but so is assessing preoperative CFS. Regarding our results, we do not recommend TMT but CFS for the estimation of OS in patients with brain metastases.

Limitations: This study was performed as a retrospective analysis of data from the neuro-oncology database. The cohort was partly heterogeneous, with some patients (13.1%) receiving more than one surgery, and therefore TMT measurements were conducted on previously operated crania. Even though TMT was then measured on the side where no previous craniotomy had been performed, there might still have occurred changes in TMT due to the influence of previous therapy (including surgery, radiation, and chemotherapy) on patients’ general condition and SMM. The study represents a selected cohort of patients who were eligible for neurosurgical treatment.

## Figures and Tables

**Figure 2 cancers-16-01660-f002:**
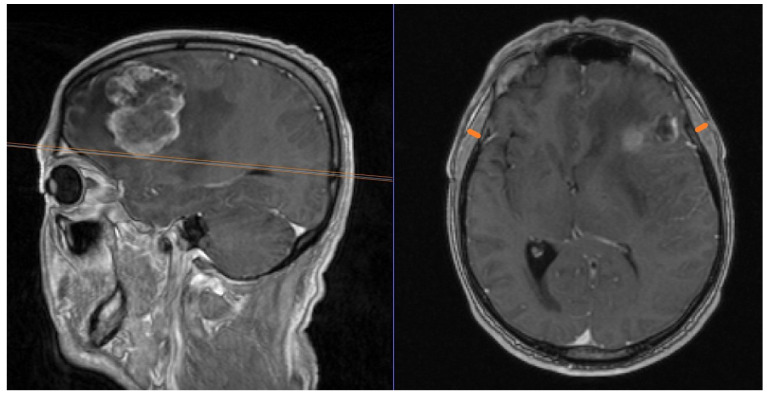
TMT measurement in a patient from our cohort with a left frontal brain metastasis. Sagittal and axial T1 contrast-enhanced images are shown. The orange lines on the axial image depict the measurement of TMT.

**Figure 3 cancers-16-01660-f003:**
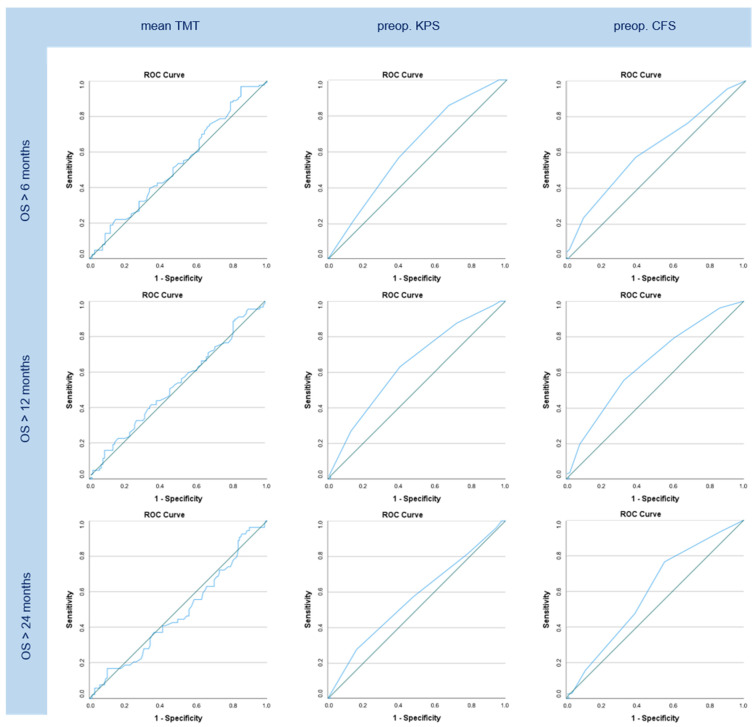
ROC graphs for OS greater than 6, 12, and 24 months regarding mean TMT, preoperative (“preop.”) KPS, and preoperative CFS. No significant AUC data are shown in the TMT column while KPS shows significant graphs in OS > 6 and > 12 months and CFS in OS > 6, > 12, and > 24 months.

**Table 1 cancers-16-01660-t001:** Most common primary tumors responsible for brain metastases in this study cohort. NSCLC = non-small-cell lung cancer; SCLC = small-cell lung cancer; ER = estrogen receptor; HER = human epidermal receptor; RCC = renal cell carcinoma; CUP = cancer of unknown primary.

Primary Tumor	*n* (%)
Lung	89 (44.7%)
NSCLC	75 (37.9%)
SCLC	12 (6.1%)
Mamma	23 (11.6%)
ER+	4 (2%)
HER2+	7 (3.5%)
ER+/HER2+	3 (1.5%)
Triple-	8 (4%)
RCC	5 (2.5%)
Colon	8 (4%)
CUP	6 (3%)
Other	41 (20.6%)

**Table 2 cancers-16-01660-t002:** Mean TMT in mm for each score of KPS assessed preoperatively (“preop.”), postoperatively (“postop.”), and in the follow-up visit (“FU”). TMT data are left blank when no patients showed the corresponding KPS.

KPS Preop. (*n*)	TMT (95CI)	KPS Postop. (*n*)	TMT (95CI)	KPS FU (*n*)	TMT (95CI)
10		10		10	
20		20 (1)		20	
30		30		30 (2)	6.1
40 (1)		40 (2)	7.4	40 (2)	8.0
50 (2)	5.6	50 (5)	6.3 (5.2–7.4)	50 (4)	7.4 (3.6–11.3)
60 (6)	6.5 (5.2–7.9)	60 (16)	6.8 (6.1–7.6)	60 (10)	7.5 (6.2–8.8)
70 (32)	7.1 (6.7–7.6)	70 (13)	7.2 (6.3–8.1)	70 (12)	7.0 (6.0–8.1)
80 (58)	7.6 (7.1–8.1)	80 (28)	7.4 (6.8–8.0)	80 (23)	8.4 (7.5–9.2)
90 (61)	7.8 (7.4–8.2)	90 (94)	7.7 (7.4–8.0)	90 (47)	7.4 (7.0–7.9)
100 (39)	7.6 (7.1–8.1)	100 (39)	7.9 (7.3–8.5)	100 (45)	7.4 (7.0–7.9)

**Table 3 cancers-16-01660-t003:** Mean TMT in mm for each score of CFS assessed preoperatively (“preop.”), postoperatively (“postop.”), and in the follow-up visit (“FU”). TMT data are left blank when no patients showed the corresponding CFS.

CFS Preop. (*n*)	TMT (95CI)	CFS Postop. (*n*)	TMT (95CI)	CFS FU (*n*)	TMT (95CI)
1 (16)	7.6 (7.0–8.2)	1 (16)	7.3 (6.5–8.2)	1 (34)	7.4 (6.8–7.9)
2 (42)	7.6 (7.0–8.1)	2 (76)	7.9 (7.5–8.3)	2 (36)	7.6 (7.1–8.2)
3 (51)	7.7 (7.3–8.1)	3 (50)	7.6 (7.1–8.1)	3 (27)	7.3 (6.8–7.8)
4 (62)	7.8 (7.3–8.2)	4 (26)	7.2 (6.6–7.8)	4 (26)	8.1 (7.3–8.9)
5 (22)	6.6 (5.9–7.2)	5 (15)	7.4 (6.7–8.2)	5 (11)	7.7 (6.5–8.9)
6 (3)	6.1 (4.9–7.3)	6 (8)	6.1 (5.4–6.8)	6 (4)	6.4 (4.0–8.9)
7 (3)	8.6 (4.1–13.2)	7(5)	7.6 (4.1–11.1)	7 (1)	
8		8 (3)	6.7 (2.4–11.0)	8 (6)	7.6 (5.4–9.8)
9		9		9	

## Data Availability

The raw data supporting the conclusions of this article will be made available by the authors on request.

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
