# Peer review of "Temporal Muscle Thickness Compared to Functional Scales as a Prognostic Parameter in Patients with Brain Metastases"

_cancers, 2024, doi:10.3390/cancers16091660_

Round 1
Reviewer 1 Report
Comments and Suggestions for Authors
In this retrospective account, temporal muscle thickness (TMS) and survival following resection of one to 3 brain metastases are compared to functional indices pre and postoperatively. No significant prognostic relevance of TMS could be confirmed, as postulated elsewhere. This is interesting. Presentation and language are appropriate.
A few aspects should be improved.
The retrospective nature of the analysis is mentioned only at the very end of the manuscript under limitations. This should be clearly stated in Methods, and whether TMS and functional indices were calculated retrospectively or had been recorded at the time of patient treatment.
There is only one patient included with resection of 3 metastases. It would have been reasonable to limit the analysis to resection of 1-2 metastases.
The potential effect of temporal craniotomies on postoperative TMS is unclear, or better how it was dealt with. In Methods it is said:” Bilateral measurements of TMT were performed and the mean was calculated.”, but later contradicted in the Discussion: “Even though TMT has then been measured on the side where no previous craniotomy had been performed there might still have occurred changes in TMT with influence by previous therapy.”
Reviewer 2 Report
Comments and Suggestions for Authors
The objective of the present study was to assess the prognostic value of the temporal muscle thickness (TMT) parameter, evaluated before and after surgery, in predicting the overall survival (OS) of neuro-oncology patients with brain metastases. Since cranial MRI is routinely deployed in the evaluation of neuro-oncology patients before and after surgery, TMT emerged in the literature as an easy-to-measure parameter for the quantitative assessment of sarcopenia (as a surrogate for skeletal muscle mass) in these patients. In the present study, which was conducted retrospectively (n=199), the authors assessed the predictive OS value of the TMT parameter in relation to the two well-established functional scales—e.g., the Karnofsky Performance Status (KPS) or the Clinical Frailty Scale (CFS)—in patients with brain metastases from lung, breast, colon and renal primary tumors or with intracranial processes from unknown primaries. TMT cut-off values were determined for each patient by receiver operating characteristic (ROC) and area under the curve (AUC) analyses and further compared to the KPS and CFS values from the preoperative visit, postoperatively before the patient was discharged, and at the first follow-up visit three to six months after surgery. The key findings of the present study were that: (i) no predictive value on OS could be proven for preoperative TMT, neither in Cox regression or in ROC analysis, (ii) no significant association could be confirmed between TMT and the KPS/CFS functional scores that were assessed before surgery and at follow-up, and (iii) while preoperative and follow-up scores demonstrated no correlation with TMT, postoperative scores assessed right before patient discharge did significantly correlate with TMT, which suggests a higher perioperative vulnerability for patients with sarcopenia. The authors conclude that TMT does not prove its usefulness as a prognostic tool for OS in patients with resected brain metastases and recommended that a functional score, such as CFS, should rather be used instead for estimation of OS in this patient population.
I found the study well written, with a sound methodology and clear conclusions. The conclusions of the study appear to be supported well by the evidence presented by the authors while the limitations of their study were also clearly spelled out. I recommend the publication of this manuscript in its present form.
Author Response
Response to Reviewer 2
Thank you very much for taking the time to evaluate the meaningfulness of our manuscript. Your comments are very much appreciated and we are honored to receive your positive response and recommendation to the editor to publish the manuscript in its present form.
We are grateful for your support.
Kind regards
Dr. J. Klingenschmid